# Expression and Functional Analysis of Two Cytochrome P450 Monooxygenase Genes and a UDP-Glycosyltransferase Gene Linked with Thiamethoxam Resistance in the Colorado Potato Beetle

**DOI:** 10.3390/insects15080559

**Published:** 2024-07-23

**Authors:** Yaqi Wang, Yitong Tian, Dongdi Zhou, Jiayi Fang, Jingwei Cao, Chengcheng Shi, Yixuan Lei, Kaiyun Fu, Wenchao Guo, Weihua Jiang

**Affiliations:** 1Department of Entomology, College of Plant Protection, Nanjing Agricultural University/Key Laboratory of Integrated Management of Crop Disease and Pests, Ministry of Education/Key Laboratory of Integrated Pest Management on Crops in East China, Ministry of Agriculture, Nanjing 210095, China; a18235860329@163.com (Y.W.);; 2China State Farms Ecnomic Development Center/South Subtropical Crops Center Ministry of Agricultureand Rural Affairs of the People’s Republic of China, Beijing 100122, China; 15690563263@163.com; 3Institute of Plant Protection, Xinjiang Academy of Agricultural Sciences/Key Laboratory of Integrated Pest Management on Crops in Northwestern Oasis, Ministry of Agriculture/Xinjiang Key Laboratory of Agricultural Biosafety, Urumqi 830091, China; fukaiyun000@foxmail.com (K.F.); gwc1966@163.com (W.G.)

**Keywords:** *Leptinotarsa decemlineata*, thiamethoxam, resistance, cytochrome P450, UDP-glycosyltransferase, RNA interference

## Abstract

**Simple Summary:**

Several differentially expressed genes encoding cytochrome P450 monooxygenases (P450s) and UDP-glycosyltransferases (UGTs), namely, *CYP9Z140*, *CYP9AY1*, and *UGT321AP1*, were screened and verified between thiamethoxam*-*susceptible and resistant populations of *Leptinotarsa decemlineata* (Say). The expression of the three genes was significantly enhanced after exposure to thiamethoxam. RNA interference of three genes increased mortality of test adults following thiamethoxam treatment. The findings reveal that the three genes have roles in the thiamethoxam resistance of *L. decemlineata*.

**Abstract:**

Cytochrome P450 monooxygenases (P450s) and UDP-glycosyltransferases (UGTs) are involved in the evolution of insecticide resistance. *Leptinotarsa decemlineata* (Say), the Colorado potato beetle (CPB), is a notorious insect that has developed resistance to various insecticides including neonicotinoids. This study investigated whether the differentially expressed P450 genes *CYP9Z140* and *CYP9AY1* and UGT gene *UGT321AP1*, found in our transcriptome results, conferred resistance to thiamethoxam in *L. decemlineata*. Resistance monitoring showed that the sampled field populations of *L. decemlineata* adults collected from Urumqi City and Qapqal, Jimsar, and Mulei Counties of Xinjiang in 2021–2023 developed low levels of resistance to thiamethoxam with resistance ratios ranging from 6.66- to 9.52-fold. Expression analyses indicated that *CYP9Z140*, *CYP9AY1*, and *UGT321AP1* were significantly upregulated in thiamethoxam-resistant populations compared with susceptible populations. The expression of all three genes also increased significantly after thiamethoxam treatment compared with the control. Spatiotemporal expression patterns showed that the highest expression of *CYP9Z140* and *CYP9AY1* occurred in pupae and the midgut, whereas *UGT321AP1* was highly expressed in adults and Malpighian tubules. Knocking down all three genes individually or simultaneously using RNA interference increased the sensitivity of adult *L. decemlineata* to thiamethoxam. These results suggest that overexpression of *CYP9Z140*, *CYP9AY1,* and *UGT321AP1* contributes to the development of thiamethoxam resistance in *L. decemlineata* and provides a scientific basis for improving new resistance management of CPB.

## 1. Introduction

The Colorado potato beetle (CPB)*, Leptinotarsa decemlineata* Say (Coleoptera: Chrysomelidae), is a notorious insect pest of solanaceous crops causing considerable economic losses. In China, the beetle is mainly distributed in the potato-growing areas north of Tianshan in Xinjiang and has spread to northeast China in recent years, posing a serious threat to potato production [1]. Currently, the application of various insecticides is still the most effective way to control CPB. The neonicotinoid agent thiamethoxam has been commonly applied for CPB in Xinjiang for nearly two decades; however, such excessive reliance has led inevitably to resistance developing in local CPB populations [2,3].

Insects develop insecticide resistance typically by decreased target sensitivity and enhanced metabolic detoxification. Mutations in nicotinic acetylcholine receptor (nAChR) subunits α1, α3, and β1 confer resistance to the neonicotinoid insecticide imidacloprid against *Nilaparvata lugens* and *Aphis gossypii* [4,5], whereas downregulation of the nAChR subunits Ldα1, Ldα3, Ldα8, and Ldβ1 from *L. decemlineata* is involved in thiamethoxam tolerance [6,7,8]. In addition, research has shown that resistance to neonicotinoids was commonly related to the enhanced activity of detoxification enzymes, particularly cytochrome P450 monooxygenases (P450s) [9]. Whole-genome sequence analysis and simultaneous examination of the expression of multiple genes revealed that P450 gene upregulation in insecticide-resistant strains resulting from the evolutionary plasticity of P450 was common in many species [10,11]; for example, overexpressed P450 genes involved in insect resistance to imidacloprid and/or thiamethoxam include *CYP6CM1* and *CYP6DB3* in *Bemisia tabaci*; *CYP6ER1* and *CYP6AY1* in *N. lugens*; *CYP6FV12* in *Bradysia odoriphaga*; and *CYP6CY14* and *CYP6DA1* in *A. gossypii* [12,13,14,15,16,17]. Other overexpressed P450 genes have also been found in imidacloprid-resistant beetles. Zhu et al. [18] reported 41 P450 genes that showed significantly higher expression in imidacloprid-resistant strains of *L. decemlineata* compared with sensitive populations. Follow-up studies identified a series of upregulated P450 genes, including *CYP9Z26*, *CYP6BQ5*, *CYP4Q3*, *CYP9Z25*, *CYP9Z29*, *CYP6BJ*^a/b^, *CYP6BJ1v1*, and *CYP6K1* [19,20,21,22].

In addition to P450s, as key phase II enzymes in detoxification, insect uridine diphosphate glycosyltransferases (UGTs) have also received attention in insecticide resistance research. For example, the midgut-specific overexpression of *UGT341A4, UGT344B49*, and *UGT344M2* significantly increased insensitivity to cyantraniliprole in *A. gossypii* [23], whereas the upregulated expression of *FoUGT466B1, FoUGT468A3*, and *FoUGT468A4* contributed to spinosad resistance in *Frankliniella occidentalis* [24]. *UGT352A5* was also reported to be responsible for conferring thiamethoxam resistance in *B. tabaci* [25], while Kaplanoglu et al. [22] found that the overexpression of *UGT2* was related to imidacloprid resistance in resistant *L. decemlineata*.

Recently, RNA interference (RNAi) has become a novel pest control technology with high specificity, selectivity, and safety. A major breakthrough has been made in biopesticides based on RNAi, with the registration of Ledprona, a dsRNA that targets the proteasome subunit beta type-5 (*PSMB5*) gene of CPB, by the United States in 2023 [26]. Therefore, screening detoxification enzyme genes related to resistance will also help in the development of this new biopesticide.

Studies indicate that different insect species and even different populations of the same insect have different metabolic resistance mechanisms to the same insecticide. However, there is limited information about which genes are involved in the molecular metabolic mechanism of resistance to thiamethoxam in CPB. In this study, the main goal was to uncover the role of P450 and UGT genes in the thiamethoxam resistance of *L. decemlineata*. Thus, transcriptome analysis was performed to screen genes encoding detoxifying enzymes that were differentially expressed between thiamethoxam-resistant and sensitive CPB populations in Xinjiang. The expression of two upregulated P450 CYP9e2-like genes (*CYP9Z140* and *CYP9AY1*) and one UGT gene (*UGT321AP1*) was further verified and analyzed in different field populations, stages, and tissues, and in response to thiamethoxam via quantitative real-time PCR (RT-qPCR). RNAi was then used to suppress the expression of these genes to explore their roles in thiamethoxam resistance. The results provided a basis for better understanding the molecular mechanisms of the metabolic resistance of *L. decemlineata* to neonicotinoid insecticides.

## 2. Materials and Methods

### 2.1. Insects

Ten CPB populations were collected from different potato fields of Qapqal County (QPQLZ and QPQLB), Mulei County (ML), Jimusa County (JMSL, JMST, JMSQ, JMSD1 and JMSD2), and Urumqi City (URMQY and URMQA) in Xinjiang from June to August in 2021, 2022, and 2023 (Table 1). CPBs were fed with potato leaves and kept in a rearing room at 26 ± 1 °C, 50–60% relative humidity, and 16 h/8 h light/dark cycle. The potato variety is Xisen No. 6 and was provided by the Integrated Testing Farm of the Xinjiang Academy of Agricultural Sciences. Adults (a mix of females and males, at least 7 d after emergence) were selected for subsequent experiments.

### 2.2. Bioassay

The contact toxicity of thiamethoxam in CPB adults was assayed using a topical application method. Thiamethoxam (97% powder, Jiangsu Bangsheng Biotechnology Co., Ltd., Huaian, China) was diluted to at least five different concentrations with analytical-grade acetone to result in a 10–100% mortality range of test insects. Ten adults were treated individually with 1.1 μL of insecticide solution or acetone as control, which was applied to their ventral area using a microapplicator (Hamilton Company, Reno, NV, USA), and then added to Petri dishes (9 cm in diameter and 1.5 cm in height) containing fresh potato leaves and maintained under the conditions described above. Each treatment had three replicates. The standard reference for dead beetles was based on Liu et al. [2], and beetle mortality was recorded after 72 h.

### 2.3. RNA-Sequencing Data Analysis

Twelve adults (three beetles for each repetition) of each population, including a thiamethoxam-susceptible population and two resistant populations with low-level resistance to thiamethoxam, were sent on dry ice to Biomarker Technologies Co., Ltd. (Beijing, China), for RNA extraction, cDNA library construction, and RNA sequencing. Sequencing was performed on an Illumina Novaseq 6000 platform (company, city, country) using a 150 bp paired-end sequencing strategy. The clean reads were aligned to the reference genome of *L. decemlineata* from the relevant genome website (https://www.ncbi.nlm.nih.gov/datasets/genome/GCF_000500325.1/ (accessed on 31 October 2021). Differential expression levels between susceptible and resistant populations were analyzed using the DESeq2 R package (1.20.0), based on fragments per kilobase per million (FPKM). The false discovery rate (FDR) was used to identify the threshold of the *p*-value in multiple tests to compute the significant difference. Genes with an absolute value of log_2_Fold Change > 1 and FDR core < 0.05 found by DESeq2 were considered to be differentially expressed.

The cloud blast feature in Blast2GO v.2 software was used to annotate the transcripts by comparing the sequences with the arthropod non-redundant protein database with a Blast expectation value (e-value) of 1.0 × 10^–5^ as a cutoff. Gene Ontology (GO) enrichment analysis was performed using Perl script by plotting the GO information of the differentially expressed genes (DEGs) retrieved from Blast2GO against all GOs from the *L. decemlineata* genome data. The obtained annotation was enriched and refined using TopGo (R package). Kyoto Encyclopedia of Genes and Genomes (KEGG) pathways were assigned to the assembled sequences by Perl script.

### 2.4. Sequence and Phylogenetic Analysis

P450 and UGT genes identified from the transcriptome and genome data of *L. decemlineata* were cloned and verified by reverse transcription PCR (RT-PCR). Total RNA was extracted from a mixture of eggs, first- to fourth-instar larvae, pupae, and adults of *L. decemlineata*. Multiple alignments of sequences were performed using GeneDoc EXE, and the structural domains were detected based on comparison with other identified sequences. The theoretical isoelectric points (pIs) and molecular weights (Mws) were analyzed by ExPASy (https://web.expasy.org/protparam/ (accessed on 15 October 2022)). MEGA 7 was utilized to construct the phylogenetic trees via the neighbor-joining method with 1000 bootstrap replications based on the amino acid sequences of CYP9e2 and UGT genes from other insects acquired through similarity searches of the NCBI database. The three verified genes were named by the P450 (David R. Nelson, Department of Molecular Sciences, University of Tennessee, Memphis, TN, USA) and UGT nomenclature committees (https://labs.wsu.edu/ugt/ (accessed on 31 March 2022) as *CYP9Z140*, *CYP9AY1*, and *UGT321AP1*, respectively.

### 2.5. Preparation of Samples for Expression Analysis

Three adults were sampled from each population from the eight sample sites in Xinjiang (detailed in Table 1) to determine the expression difference in three candidate genes among different CPB field populations. In order to extract RNA of similar concentration, the number of the beetle samples at different stages of development collected was different to make their quality similar. We collected 30 eggs (E), 30 1st-instar larvae (L1), 20 2nd-instar larvae (L2), 10 3rd-instar larvae (L3), as well as 3 4th-instar larvae (L4), 3 pupae (P) and 3 adults (A), from the URMQA population to examine the stage-specific expression of three genes. To compare the tissue expression of candidate genes, the foreguts, midguts, hindguts, Malpighian tubules, fat bodies, head, thorax, and abdomen were dissected from five adults of URMQA, respectively. Three adults were sampled from the survivors of URMQA treated with either LD_50_ of thiamethoxam, or acetone treatment (as control) for 72 h was used to determine the inducible expression profiles of three genes. The sample size above is taken as a biological replication, and each treatment (population) had three biological replicates. All samples were frozen quickly in liquid nitrogen and stored at –80 °C until use.

### 2.6. Total RNA Isolation and cDNA Synthesis

Total RNA from the above-mentioned samples was isolated using Yfx Total RNA Extraction Reagent (Yi Fei Xue Biotechnology Co., Ltd., Nanjing, China), following the manufacturer’s protocol. The concentration of the RNA samples was analyzed on a NanoDrop 1000 spectrophotometer (Thermo Fisher Scientific, Waltham, MA, USA). The first-strand cDNA was then synthesized by using a PrimeScript RT reagent kit (TaKaRa Biotechnology Co., Ltd., Dalian, China).

### 2.7. Real-Time Quantitative PCR

The transcript levels of *CYP9Z140*, *CYP9AY1*, and *UGT321AP1* were determined using a Biosystems 7500 Real-time PCR System (Applied Biosystems Inc., Foster City, CA, USA) with ChamQTM SYBR qPCR Master Mix (Vazyme Biotech Co., Ltd., Nanjing, China). qPCR reaction mixtures comprised 10 μL SYBR Green, 0.4 μL of each primer (10 μmol/L), 1 μL cDNA template (300 ng/μL), and 8.2 μL RNase-free water. The reaction involved the following steps: initial step at 95 °C for 30 s, followed by 40 cycles of 95 °C for 5 s and 60 °C for 34 s. The primers used are detailed in Table 2. There were three independent biological replicates for each qPCR experiment. The relative expression of the target genes was calculated according to the 2^−ΔΔCT^ method [27], with ribosomal protein L4(*RPL4*) and translation elongation factor 1α*(EF1a*) as reference genes, based on Zhu et al. [18].

### 2.8. RNA Interference

*CYP9Z140*-double-stranded (ds)RNA, *CYP9AY1*-dsRNA, *UGT321AP1*-dsRNA, and *GFP*-dsRNA were expressed using *Escherichia coli* HT115 (DE3) competent cells lacking RNase III, following the methods of Qu et al. [6] and Shi et al. [8].

Potato leaves of similar size were dipped in bacterial solutions containing ds*CYP9Z140*, ds*CYP9AY1*, ds*UGT321AP1*, and ds*GFP* (as control) for 30 min, and then placed in plastic feeding chambers with 17 cm in length, 11.7 cm in width, and 5 cm in height after air drying. Adult CPB collected from URMQA were carefully transferred into each chamber containing the treated leaves. Thirty beetles were used for each treatment, and all treatments were replicated 6 times. A fresh supply of treated potato leaves was provided daily. To silence the three genes simultaneously, the beetles were fed a mixture of dsRNA of the three genes at a 1:1:1 ratio.

After 6 d of continuous feeding on treated leaves, four replicates from each treatment group (12 adults) were used to extract total RNA for measuring the expression levels of the target genes, as detailed above. The remaining beetles from each treatment group were used to determine the susceptibility to thiamethoxam. Adults were treated with a median lethal dose (LD_50_) of thiamethoxam (0.2963 μg/adult), with the same amount of acetone used as a control. Each treatment was repeated four times with 15–20 adults each. The number of dead beetles in each group was then recorded, as described above.

### 2.9. Statistical Analysis

Bioassay data were corrected for control mortality by Abott’s formula. Median lethal doses (LD_50_) and 95% fiducial limits (FLs) were estimated via PoloPlus 2.00 software (Leora Software, Berkeley, CA, USA). The resistance ratio (RR) was calculated by dividing the LD_50_ value of the field population by the LD_50_ value of the susceptible population and was quantified according to Shi et al. [3]. The quantitative data of three genes and mortality of test beetles exposed to thiamethoxam after RNAi were expressed as the mean ± standard error (SE) from at least three biological replicates. Data on transcriptome validation and inducible expression were analyzed to compare the difference between the two treatments using Student’s *t*-test. The remaining data were analyzed by one-way analysis of variance (ANOVA) followed by Tukey’s multiple comparison tests. Statistical analysis was carried out using GraphPad Prism 8.02 and SPSS statistics (IBM SPSS Statistics 27 software, Chicago, IL, USA). Statistical differences were considered significant at *p* < 0.05.

## 3. Results

### 3.1. Resistance Levels of L. decemlineata Populations to Thiamethoxam

The sensitivity to thiamethoxam of *L. decemlineata* field populations collected from Xinjiang in 2021, 2022, and 2023 were assayed by topical application (Table 3). The URMQY population was considered to be relatively sensitive. In 2021, the ML population remained sensitive to thiamethoxam, with an RR of 2.18-fold, whereas the URMQA population showed decreased susceptibility, with an RR of 3.04-fold. The JMSL and QPQLZ populations developed 7.18- and 8.33-fold low levels of resistance to thiamethoxam, respectively. In 2022, the URMQA and JMST populations had low levels of resistance and decreased susceptibility, with RRs of 9.53- and 3.23-fold, respectively. In 2023, Both ML and JMSD1 developed low resistance, with RRs of 7.42- and 6.66-fold, respectively. The JMSQ, URMQA, and JMSD2 populations showed decreased susceptibility with RRs ranging from 3.04-fold to 4.63-fold. QPQLB remained sensitive to thiamethoxam throughout the study period. The results of resistance monitoring to thiamethoxam may provide the basis for effective control of *L. decemlineata*.

### 3.2. Transcriptome Analysis

Illumina short-read sequences from mRNAs isolated from the URMQY, QPQLZ, and JMSL populations were compiled into a transcriptome, generating 26,114,887, 21,000,522, and 24,582,542 usable reads, respectively. The percentage of Q30 bases was 93.39% and above, and the GC content of each population ranged from 40.52% to 41.23% (Table 4).

Analysis of the log-fold change in expression of genes that were significantly up- or downregulated in all samples revealed 813 DEGs between the URMQY and JMSL populations, of which 263 (32.35%) were upregulated and 550 (67.65%) were downregulated. In addition, there were 883 DEGs detected between the URMQY and QPQLZ populations, of which 254 (28.77%) were upregulated and 629 (71.23%) were downregulated.

GO analyses indicated that the annotated DEGs could be divided into three different categories as follows: biological process (BP); cellular component (CC); and molecular function (MF) (Figure 1). In each of these three main categories, the terms “metabolic process”, “cellular process”, “cell part”, “binding”, and “catalytic activity” were the most dominant. The top 20 enriched KEGG pathways were mainly linked with the metabolism of xenobiotics (Figure 2).

Several classes of detoxifying enzymes involved in enzymatic detoxification mechanisms were upregulated between susceptible and resistant populations (Table 5). Two P450 genes with a fold change > 2 and FDR score < 0.0001 (ID: 111518298 and 111508919) and two UGT genes were used for subsequent analysis.

RT-qPCR was performed to confirm the transcript expression obtained from the RNA-sequencing data. The expression levels of *CYP9Z140* (ID: 111518298) and *CYP9AY1* (ID: 111508919) in the QPQLZ population and *UGT321AP1* (ID: 111517685) in the JMSL population were significantly enhanced by 1.92, 2.04, and 36.4 times, respectively (*p* < 0.05), compared with the URMQY population, which was consistent with the transcriptome results (Figure 3). However, *UGT324BR1* (ID: 111518183) was not upregulated significantly in the QPQLZ population compared with the URMQY population and, thus, was not considered as a candidate gene for follow-up studies.

### 3.3. Gene Structure and Phylogenetic Analysis

The structural features of the two P450 genes (CYP9Z140 and CYP9AY1) are illustrated in Figure 4A. The full-length genes contained a 1578 bp open reading frame (ORF) encoding 525 amino acid residues. The theoretical isoelectric points (pIs) were 8.95 and 5.73, and the molecular weights (Mw) were 60.435 and 60.727 kDa, respectively. Conserved domains in the genes are common to cytochrome P450s and include the C-helix motif (WxxxR), I-helix motif (GxE/DTT/S), K-helix motif (ExLR), the conserved amino acid sequence PxxFxP motif, and the heme-binding motif (PFxxGxxxCxG). Structural features of the UGT gene UGT321AP1 are shown in Figure 5A. The full-length cDNA of UGT321AP1 encoded 517 amino acids. Its pI and Mw were 6.88 and 59.20 kDa, respectively. Similar to UGTs in other insects, the signal peptide of UGT321AP1 was found at the N terminus, and the signature motif ([FVA]-[LIVMF]-[TS]-[HQ]-[SGAC]-G-x(2)-[STG]-x(2)-[DE]-x(6)-P-[LIVMFA]–[LIVMFA]-x(2)-P-[LMVFIQ]-x(2)-[DE]-Q) was situated in the middle of the C-terminal domain; two sugar donor-binding site domains (DBR1 and DBR2) were also predicted for the amino acid sequences. A hydrophobic transmembrane domain containing ~29 hydrophobic amino acid residues was found at the C terminus.

The phylogenetic relationships among the three genes from *L. decemlineata* and related P450s and UGTs from other insects are shown in Figure 4B and Figure 5B. The translated proteins of P450s shared the highest amino acid sequence identity with the CYP9 subfamily of *Tenebrio molitor* and *Diabrotica virgifera virgifera* from Coleoptera. UGT321AP*1* was clustered in the branch of *Tribolium castaneum* (Coleoptera).

### 3.4. Expression Analysis of P450 and UGT Genes

RT-qPCR was used to analyze and compare the expression levels of the three genes in CPB adults from different field populations (Figure 6). In 2021, the expression levels of *CYP9Z140* in the QPQLZ and URMQA populations, *CYP9AY1* in the QPQLZ and JMSL populations, and UGT321AP*1* in the URMQA and JMSL populations increased significantly by 1.93 and 3.15 times (F = 29.37, df = 4, 10; *p* < 0.05), 1.62 and 1.71 times (F = 7.83, df = 4, 10; *p* < 0.05), and 12.78 and 21.06 times (F = 29.124, df = 4, 10; *p* < 0.05), respectively, compared with the URMQY population. In 2022, the transcript levels of *CYP9Z140* and UGT321AP*1* were significantly increased in the URMQA population by 1.73 times (F = 18.149, df = 2, 6; *p* < 0.05) and 23.30 times (F = 36.933, df = 2, 6; *p* < 0.05), respectively. In 2023, *CYP9Z140* was overexpressed in the URMQA, JMSD1, JMSD2, and ML populations by 1.71-, 2.01-, 1.69-, and 2.11-fold, respectively (F = 5.771, df = 6,14; *p* < 0.05). *CYP9AY1* expression in the URMQA and JMSD1 populations increased by 1.84 and 6.25 times, respectively (F = 22.586, df = 6,14; *p* < 0.05), whereas that of UGT321AP*1* was upregulated by 2.47, 1.66, 5.85, 3.16, 1.65, and 3.07 times (F = 36.299, df = 6,14, *p* < 0.05) in the JMSD1, JMSD2, URMQA, ML, JMSQ, and QPQLB populations, respectively. There were no other significant differences in the expression of the genes between the URMQY population and the other field populations across any of the study years.

To study the effect of neonicotinoids on the expression of CYP9Z140, CYP9AY1, and UGT321AP1, the mRNA levels in URMQY adults treated with thiamethoxam LD_50_ for 72 h were determined and analyzed by RT-qPCR (Figure 7). Thiamethoxam treatment significantly increased the expression of all three genes by 2.31- (t = 3.217, *p* = 0.0324), 3.03- (t = 5.446, *p* = 0.0055), and 5.01-fold (t = 5.796, *p* = 0.0044), respectively, compared with the control.

To identify the tissue-specific detoxification gene(s) that accounts for thiamethoxam resistance, the expression patterns of *CYP9Z140*, *CYP9AY1*, and *UGT321AP1* were analyzed by RT-qPCR in the seven developmental stages (eggs, first to fourth instar larvae, pupae, and adults), different tissues (foregut, midgut, hindgut, fat body, and Malpighian tubules), and different body parts (head, thorax, and abdomen) of *L. decemlineata*. *CYP9Z140*, *CYP9AY1*, and *UGT321AP1* were transcribed throughout all developmental stages of *L. decemlineata*. In the developmental stages, a similar pattern was detected between *CYP9Z140* and *CYP9AY1*, whereas there were clear differences in the developmental expression levels of *UGT321AP1* (Figure 8A). The expression of the two CYP450 genes was highest in pupae, followed by adult and fourth-instar larvae, and was lowest in the egg stage. The expression levels of *CYP9Z140* and *CYP9AY1* in the pupae, fourth-instar larvae, and adults were 31.10, 14.81, and 14.45 (F = 331.277, df = 6,14; *p* < 0.05) and 20.79, 13.35, and 15.84 times (F = 67.698, df = 6,14; *p* < 0.05) higher than those in the egg stage, respectively. *UGT321AP1* expression was highest in the second-instar larvae and adults, and lowest in the first-, second-, and fourth-instar larvae and pupae (F = 41.726, df = 6,14; *p* < 0.05). Tissue expression patterns showed that expression of the two P450 genes was highest in the midgut, whereas that of *UGT321AP1* was highest in Malpighian tubules (Figure 8B). The expression of *UGT321AP1* in the head and thorax was higher than that in the abdomen, with no significant difference in the expression of the two P450s in the different body parts of CPB adults (Figure 8C).

### 3.5. RNAi Effects of CYP9Z140, CYP9AY1, and UGT321AP1 on L. decemlineata

RT-qPCR was used to detect the expression levels of *CYP9Z140*, *CYP9AY1*, and *U*GT321AP*1* in beetles fed on individual dsRNA and a mixture targeting three genes for 6 d. The results showed that the expression of *CYP9Z140*, *CYP9AY1*, and UGT321AP*1* significantly reduced by 88.86% and 87.28% (F = 143.440, df = 2,6 *p* < 0.05), 79.37% and 95.08% (F = 184.814, df = 2,6; *p* < 0.05), and 84.23% and 78.27% (F = 41.105, df = 2,6; *p* < 0.05), respectively, compared with ds*GFP* (control) after individual and simultaneous RNAi (Figure 9A). Thiamethoxam at LD_50_ was also used to determine changes in the sensitivity of the treated CPB. The mortality of adults fed ds*CYP9Z140*, ds*CYP9AY1*, ds*UGT321AP1*, and a mixture of the three dsRNAs significantly increased, by 10.53% 14.55%, 13.03%, and 18.33% (F = 30.218, df = 4,15, *p* < 0.05), respectively, compared with the control group (Figure 9B).

## 4. Discussion

CPB is a species characterized by the rapid development of resistance to a variety of insecticides [28]. With the extensive application of neonicotinoids for the control of CPB in Xinjiang, it is necessary to continuously monitor such resistance. In the current study, the resistance levels to thiamethoxam of CPB from different areas of Xinjiang were investigated across three sample years (2021–2023). The LD_50_ value of the QPQLZ population to thiamethoxam (0.2592 μg⸳ adult^−1^) in our study was >0.0196 μg⸳ adult^−1^ of individuals from the same population collected in 2010 [2], while the resistance level of QPQLZ, with a RR of 8.33-fold (Table 3), was higher than the RR of 4.3-fold in the same population reported by Shi et al. [3]. The resistance level to thiamethoxam in the ML population increased from 2.18-fold in 2021 to 7.42-fold in 2023, whereas the JMST, JMSQ, and JMSD2 populations exhibited decreased susceptibility to thiamethoxam. By contrast, the URMQA population in 2022, and the JMSD1 and JMSL populations, showed low resistance. We investigated and found that Arika suspension (thiamethoxam being the main ingredient) has been applied long-term in the Jimusar (JMS) region to control CBP. This could explain why all tested populations from Jimusar showed increased levels of tolerance to thiamethoxam. In addition, it can be found that the resistance levels to thiamethoxam of CPB populations tested are low. The possible reasons are as follows: firstly, the integrated control for the beetle in the local area is rather effective, including potato field rotation, alternative use of different types of insecticides, etc., which may delay the resistance development to thiamethoxam of CPB to a certain extent. Secondly, the highly resistant populations could not be obtained because of limited sampling sites. In subsequent investigations, we will add more sampling sites to monitor the resistance dynamics of local populations.

Developments in molecular biology and genomics have led to the mechanism of insecticide resistance mediated by genes encoding detoxification enzymes becoming a hot research topic. Many studies have reported that overexpression of P450 and UGT genes can lead to resistance of pests to neonicotinoid insecticides. For example, Zhu et al. [18] revealed 41 highly expressed P450 genes in imidacloprid-resistant populations of CPB from Long Island, New York, USA. Using transcriptional analysis, Clements et al. [19,20] found that expression levels of *CYP9Z26* and *CYP96K1* in imidacloprid-resistant adults from Wisconsin, USA, were significantly increased after imidacloprid treatment. In addition, qPCR analysis showed that *CYP6K1* was also overexpressed in field populations under long-term use of neonicotinoid insecticides [29]. Kaplanoglu et al. [22] revealed that overexpression of two genes encoding detoxifying enzymes (*CYP4Q3* and *UGT2*) contributed to imidacloprid resistance in medium-level imidacloprid-resistant CPB populations. Based on these studies, it appears that overexpressed detoxification enzyme genes related to neonicotinoid resistance in CPB differ across populations with diverse resistance backgrounds. Thus, in the present study, transcriptome analysis was used to compare susceptible and resistant populations of *L. decemlineata*, revealing the DEGs *CYP9Z140*, *CYP9AY*, and *UGT321AP1*, as verified by qPCR. However, there was no evidence to suggest that these genes were directly involved in neonicotinoid resistance.

Further qPCR analysis showed that *CYP9Z140*, *CYP9AY1*, and *UGT321AP1* were overexpressed significantly in thiamethoxam-resistant adults from the QPQLZ and JMSL populations in 2021, the URMQA population in 2022, and the JMSD1 and ML populations in 2023, comparable to CYP9e2-like genes reported to be overexpressed in resistant compared with susceptible adults [3]. Therefore, we speculated that these genes were related to the resistance of *L. decemlineata* to thiamethoxam, based on the results of constitutive expression analyses of different resistant populations in Xinjiang.

Many studies have shown that insect P450 and UGT genes can be induced by insecticides. For example, *CYP6AX1* and *CYP6AY1* of *N. lugens* and *CYP6AY3v2* of *Laodelphax striatellus* were upregulated in the presence of imidacloprid [30,31]. The expression of *UGT352A4* and *UGT352A5* in the thiamethoxam-resistant *B. tabaci* strain significantly increased after thiamethoxam treatment [24]. Our study showed that thiamethoxam exposure significantly increased the expression of the three genes in URMQY adults. Similarly, the transcript level of *CYP9e2* increased 4.2-fold in *L. decemlineata* exposed to clothianidin [32]. In addition, CYP9e2-like genes are involved in insect resistance to a variety of insecticides. Oppert et al. [33] found that the expression of *CYP9e2* in susceptible populations of *T. castaneum* increased when exposed to sublethal doses of phosphine. Jiang et al. [34] reported that the relative expression level of *AcCYP9e2* in the midgut of *Apis cerana* workers was significantly higher than that of the control group after exposure to flumethrin. Using transcriptome analysis, Gao et al. [35] revealed that *CYP9e2* of *Plutella xylostella* was upregulated after treatment with chlorantraniliprole, cypermethrin, dinotefuran, indoxacarb, and spinosad. Therefore, we further speculate that *CYP9Z140*, *CYP9AY1,* and *UGT321AP1* are associated with the detoxification of thiamethoxam in *L. decemlineata*.

The specific spatiotemporal expression patterns of genes encoding detoxifying enzymes are usually linked to their protein function. The current analysis showed that *CYP9Z140*, *CYP9AY1*, and *UGT321AP1* were detected in all developmental stages and tissues of CPB, albeit with significantly different expression levels. The expression of *CYP9Z140* and *CYP9AY1* was highest in pupae and the midgut, while that of *UGT321AP1* was highest in adults and Malpighian tubules. In addition, *UGT321AP1* showed higher expression in the head and thorax than in the abdomen of CPB, whereas the expression of *CYP9Z140* and *CYP9AY1* showed no difference among the body parts. Similarly, *CYP6FV12* of *B. odoriphaga* was highly expressed in the midgut but expressed at low levels in eggs [15], and *CYP303a1* of *Drosophila melanogaster* was markedly overexpressed during the pupal stage [36]. *UGT353G2* in *B. tabaci* adults had the highest expression across different development stages [37]. However, the highest stage-specific expression of *CYP6FV12* was observed in fourth-instar nymphs of *B. odoriphaga*, and *Cyp303a1* had the highest expression in the ring gland of *D. melanogaster* [15,36]. The insect midgut and Malpighian tubules are important organs for detoxifying exogenic compounds, such as insecticides. Thus, our stage- and tissue-specific expression profiles suggested that these three genes were involved in CPB resistance to thiamethoxam and that the major detoxification action stages might occur in adults and pupae, followed by fourth-instar larvae.

Many studies have indirectly verified the roles of P450 and UGT genes in pest resistance through RNAi. For example, results from RNAi showed that *CYP6ER1* not only had a role in the resistance of *N. lugens* to imidacloprid but was also closely related to the generation of thiamethoxam and dinotefuran resistance [13,38]. In addition, the overexpressed gene *CYP6CY14* was confirmed as having an important role in the thiamethoxam resistance of *A. gossypii* [14]. The ingestion of dsRNAs for *L. decemlineata* successfully reduced the expression of *CYP9Z26* and *CYP9Z29* and increased the imidacloprid susceptibility of test beetles [21,22,28,39]. In our study, after *L. decemlineata* adults were continuously fed with bacterial solutions containing individual or mixed dsRNA of three genes for 6 d, the expression levels of the target genes and the tolerance of test beetles to thiamethoxam were significantly decreased compared with ds*GFP* treatment. The roles of CYP9e2-like genes in the insecticide resistance of insects have also been reported. Bouafoura et al. [32] found that *CYP9e2* knockdown increased the susceptibility of *L. decemlineata* to clothianidin. A cytochrome P450, *CYP9E2*, and a long non-coding RNA gene *lncRNA-2* were found to be upregulated in a Spinosad-resistant population of CPB, and knockdown of these two genes using RNAi resulted in a significant increase in spinosad sensitivity, which implies *CYP9E2* and *lncRNA-2* jointly contribute to spinosad resistance [40]. In addition, the suppression of *UGT353G2* expression by RNAi substantially increased sensitivity to multiple neonicotinoids in resistant strains of *B. tabaci,* indicating the involvement of *UGT353G2* in the neonicotinoid resistance of whitefly [36]. The current study not only confirmed overexpression of the three target genes as an important resistance mechanism to neonicotinoids, but also indicated that different populations of *L. decemlineata* had different metabolic molecular mechanisms based on the RNAi effects of *UGT321AP1*, *CYP9Z140*, and *CYP9AY1* on sensitivity to thiamethoxam. Our findings suggested that RNAi-triggered knockdown of *CYP9Z140*, *CYP9AY1*, and *UGT321AP1* resulted in an increased susceptibility to thiamethoxam in the adults of the field populations, which may provide a scientific basis for improving new management of *L. decemlineata*.

Our study results showed that two P450 genes and one UGT gene conferred resistance to thiamethoxam, indicating that thiamethoxam resistance in *L. decemlineata* develop by a complex mechanism. Thus, other detoxification genes related to the thiamethoxam resistance of CPB need to be screened and identified. Furthermore, the regulatory mechanism of *CYP9Z140*, *CYP9AY1*, and *UGT321AP1* expression remains to be elucidated.

## 5. Conclusions

In summary, using resistance monitoring, this study showed that most test populations in Xinjiang developed low resistance to thiamethoxam. The results of RT-qPCR analysis determined not only that *CYP9Z140*, *CYP9AY1*, and *UGT321AP1* were overexpressed in resistant populations but also that their expression was induced by thiamethoxam and that they were highly expressed in the midgut and Malpighian tubules. RNAi further confirmed the roles of the genes in the development of resistance to thiamethoxam against *L. decemlineata*. These results will facilitate the development of CPB resistance management strategies.

## Figures and Tables

**Figure 1 insects-15-00559-f001:**
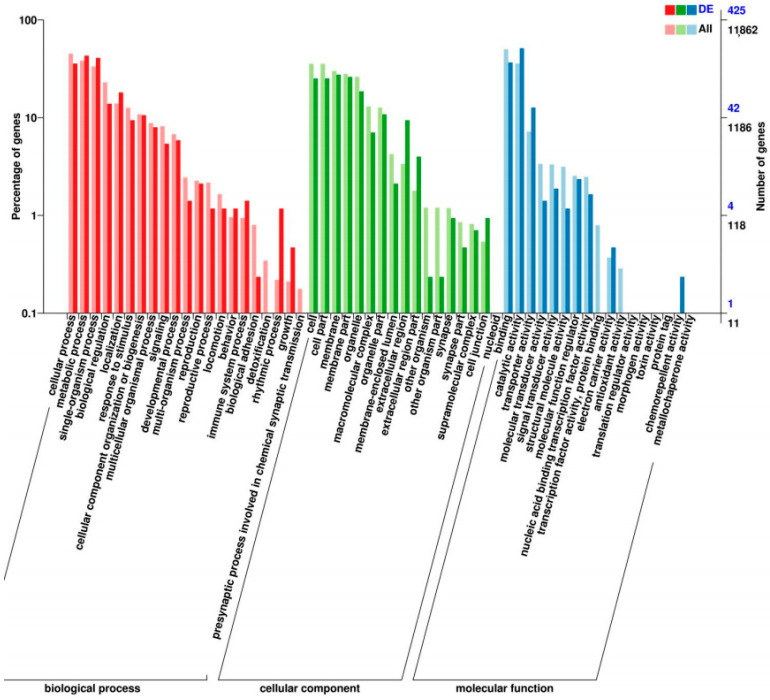
GO functional annotation of differentially expressed genes.

**Figure 2 insects-15-00559-f002:**
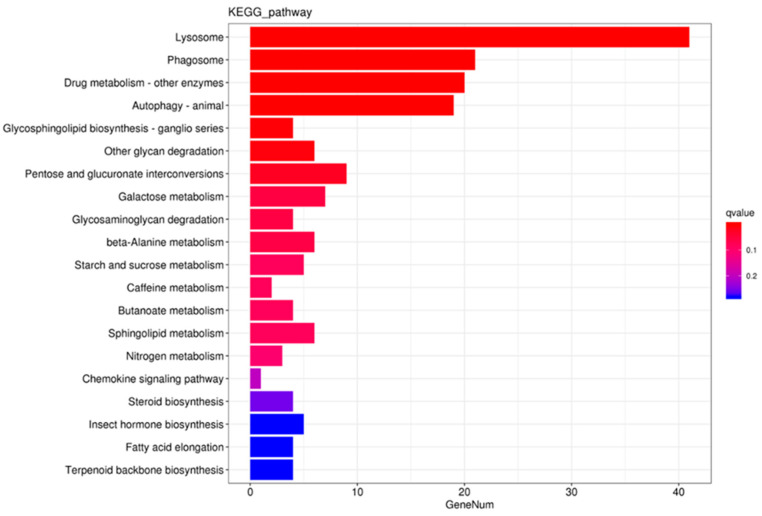
Kyoto Encyclopedia of Genes and Genomes (KEGG) enrichment histogram of differentially expressed genes.

**Figure 3 insects-15-00559-f003:**
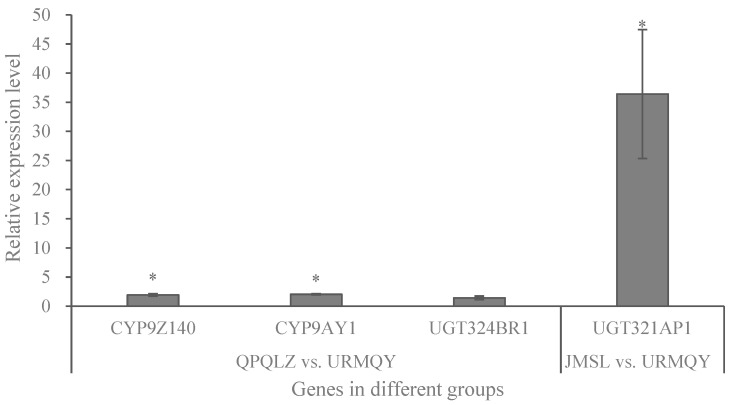
Quantitative validation of differentially expressed genes from transcriptome data. Fold increase in normalized mRNA expression levels of P450 and UGT genes in the resistant populations JMSL (collected from Jimsar Couty) or QPQLZ (collected from Qapqal County) in 2021 relative to normalized expression levels (set to one) in the susceptible population (URMQY, collected from Urumqi City). The bar labeled in each column indicates the sample mean ± SE. Asterisks (*) represent significant changes in the mRNA transcript level of each gene in qPCR results at the *p* < 0.05 level (Student’s *t*-test).

**Figure 4 insects-15-00559-f004:**
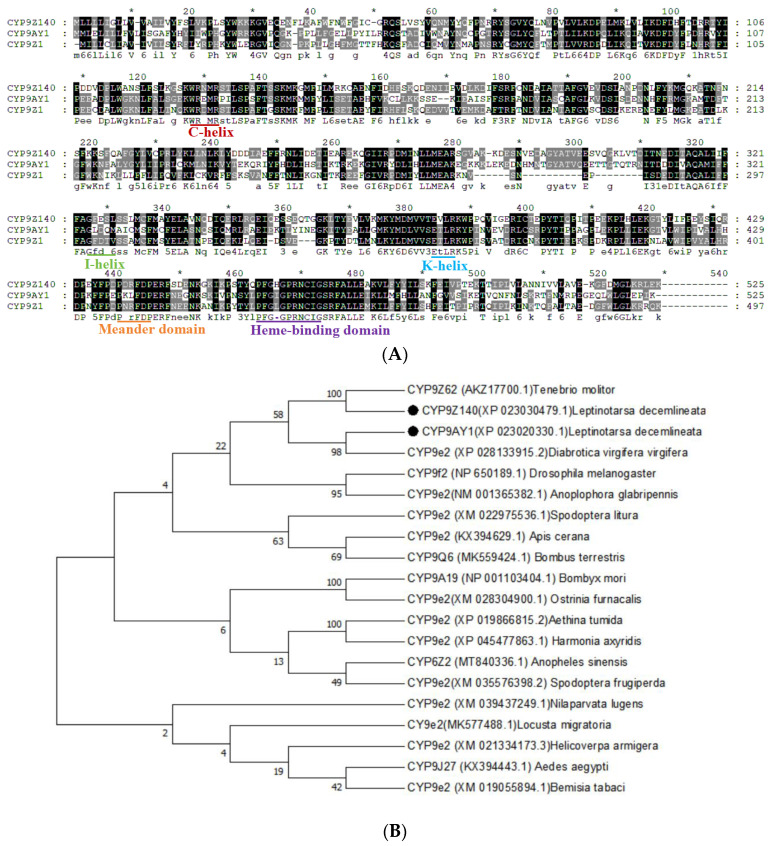
Bioinformatic analysis of two P450 genes *CYP9Z140* and *CYP9AY1* from *L. decemlineata*. (**A**) Alignment of amino acid sequences of *CYP9Z140*, *CYP9AY1,* and the related P450 gene CYP9Z1 from *Tribolium castaneum*. Conserved motifs were highlighted in the sequences, including the helix-C motif (WxxxR), the oxygen-binding motif (helix I) ([A/G] GX [E/D] T[T/S]), the helix K motif (EXXRXXP), the conserved Meander motif (PXXFXP), and the heme-binding motif (PFXXGXXXCXG). (**B**) Phylogenetic tree of CYP9Z140, CYP9AY1, and related P450s from other insects. Bootstrap values (1000 replicates) are indicated next to the branches, and GenBank accession numbers are shown in parentheses. The black dot indicates *CYP9Z140* and *CYP9AY1* in *L. decemlineata*.

**Figure 5 insects-15-00559-f005:**
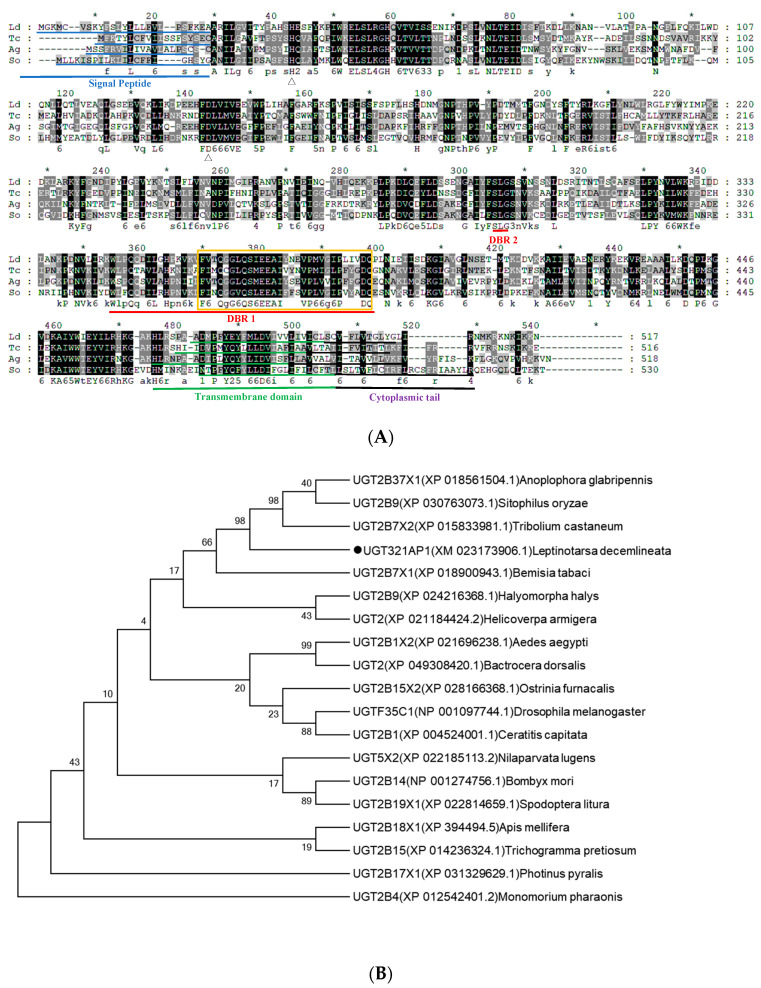
Bioinformatic analysis of UGT gene *UGT321AP1* from *L. decemlineata*. (**A**) Alignment of amino acid sequences of *UGT321AP1* and related UGT gene from *T*. *castaneum*, *Sitophilos oryzae,* and *Anoplophora glabripennis*. The signal peptides in the N terminus are shown with a blue underline. The UGT signature motif is shown with a yellow box. The transmembrane domains in the C-terminal half and cytoplasmic tail are shown in green and purple underlines. The red bars under the sequences indicate the two donor-binding regions (DBR1 and DBR2). (**B**) Phylogenetic tree of *UGT321AP1* and related UGTs from other insects. Bootstrap values (1000 replicates) are indicated next to the branches, and GenBank accession numbers are shown in parentheses. The black dot indicates *UGT321AP1* in *L. decemlineata*.

**Figure 6 insects-15-00559-f006:**
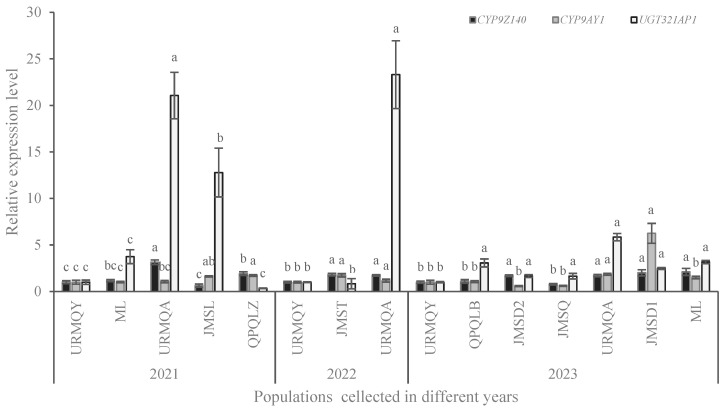
Relative expression levels of *CYP9Z140*, *CYP9AY1*, and *UGT321AP1* in different field populations of *L*. *decemlineata* in 2021, 2022, and 2023. Data are expressed as mean relative quantity ± SE. The expression levels of the three genes were normalized and calculated using *EF-1α* and *RPL4* as internal reference genes. Different lowercase letters above the bars represent significant expression differences in each gene in different populations compared to susceptible populations using one-way ANOVA followed by Tukey’s multiple comparisons (*p* < 0.05).

**Figure 7 insects-15-00559-f007:**
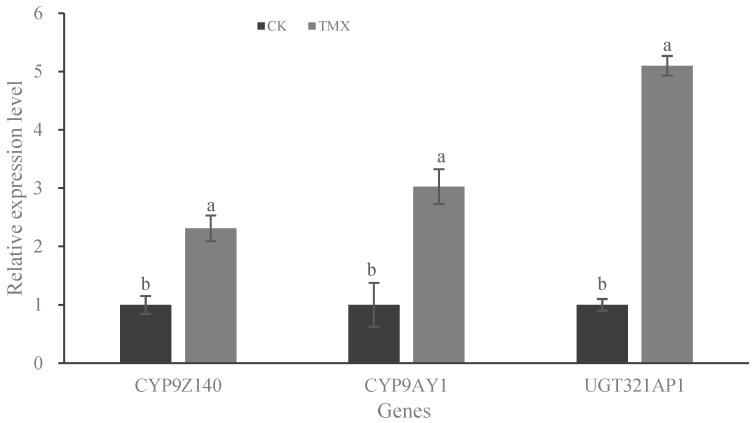
Expression levels of P450 and UGT genes in *L*. *decemlineata* adults treated with LD_50_ of thiamethoxam (TMX). The mRNA expression levels of three genes in the URMQA population exposed to acetone for 72 h were used as controls. The expression of the test genes was normalized and calculated using *EF-1α* and *RPL4* as internal reference genes. Different lowercase letters above the bars represent significant differences in mRNA levels between treatment and control for each gene by Student’s *t*-test (*n* = 3, mean relative quantity ± SE, *p* < 0.05).

**Figure 8 insects-15-00559-f008:**
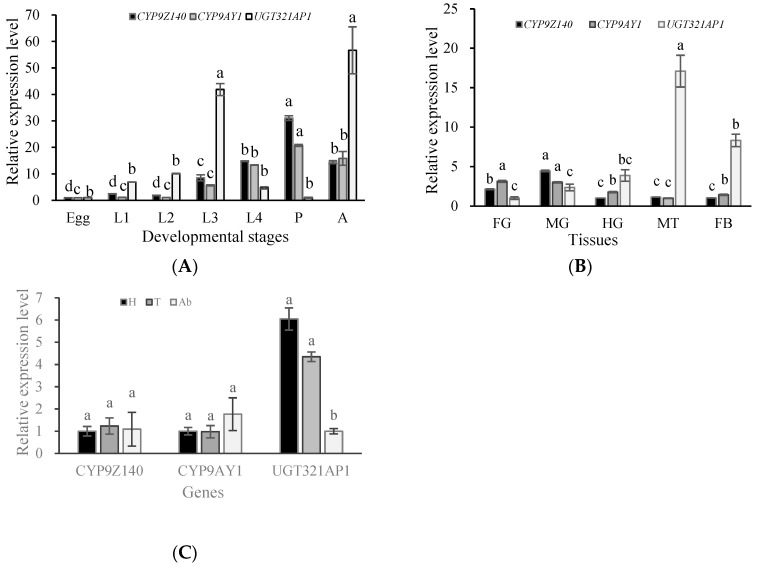
Spatiotemporal expression patterns of *CYP9Z140*, *CYP9AY1*, and *UGT321AP1* in *L. decemlineata*. (**A**) Relative expression levels of the three genes in developmental stages. L1: first-instar larva; L2: second-instar larva; L3: third-instar larva; L4: fourth-instar larva; P: pupa; A: adult. (**B**) Relative expression levels of the three genes in different tissues of *L. decemlineata* adults. FG: foregut; MG: midgut; HG: hindgut; MT: Malpighian tubule; FB: fat body. (**C**) Relative expression levels of the three genes in different parts of *L. decemlineata* adults. H: head; T: thorax; Ab: abdomen. Data are expressed as mean relative quantity ± SEM. Different lowercase letters above the bars represent significant differences in mRNA levels for each gene in different stages or tissues using one-way ANOVA followed by Tukey’s multiple comparisons (*n* = 3, *p* < 0.05).

**Figure 9 insects-15-00559-f009:**
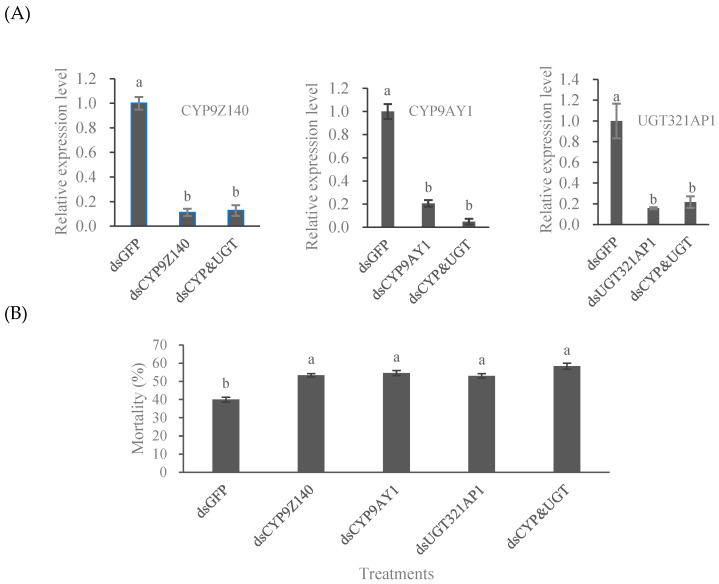
Effects of RNA interference on mRNA expression of the three genes (**A**) and on sensitivity to thiamethoxam of adult *L. decemlineata* (**B**). (**A**) Quantitative PCR analysis was used to determine the expression of *CYP9Z140*, *CYP9AY1*, and *UGT321AP1* in URMQA adults after feeding on a diet containing individual dsRNA or a mixture of dsRNA (ds*CYP9Z140*, ds*CYP9AY*, and ds*UGT321AP1*) for 6 days. The expression level obtained with dsRNA of target genes is shown relative to that obtained with ds*GFP*, which was assigned a value of 1. (**B**) Mortality was recorded for adult *L. decemlineata* exposed to thiamethoxam (0.2963 μg/adult) for 72 h after individual and simultaneous RNAi for 6 d. Adults were fed with ds*GFP* as control. All values are means + SEs of three biological replicates. Different lowercase letters above the bars represent significant differences for each treatment using one-way ANOVA followed by Tukey’s multiple comparisons (*n* = 3, *p* < 0.05).

**Table 1 insects-15-00559-t001:** Background information of *L. decemlineata* populations collected from Xinjiang.

Sampling Date	Population	Sampling Location	Co-Ordinates
June 2021	QPQLZ	Development zone of Zakuqiniulu Town, Qapqal County, Yili Prefecture	43.50 N, 81.17 E
June 2021	ML	Dongcheng Town, Mulei County, Changji Prefecture	43.83 N, 90.29 E
June 2021	JMSL	Louzhuangzi Village, Jimsar County, Changji Prefecture	43.46 N, 89.8 E
June 2021	URMQA	Anningqu Town, Xinshi District, Urumqi City	43.57 N, 87.28 E
July 2021	URMQY	Yongfeng Town, Urumqi County, Urumqi City	43.40 N, 87.19 E
July 2022	JMST	Taiping Village, Jimsar County, Changji Prefecture	43.77 N, 89.16 E
July 2022	URMQA	Anningqu town, Xinshi District, Urumqi City	43.57 N, 87.28 E
June 2023	URMQA	Anningqu Town, Xinshi District, Urumqi City	43.57 N, 87.28 E
June 2023	JMSQ	Quanzijie Town, Jimsar County, Changji Prefecture	43.83 N, 87.62 E
June 2023	ML	Dongcheng Town, Mulei County, Changji Prefecture	43.83 N, 90.29 E
July 2023	QPQLB	Development zone of Ba Town, Qapqal County, Yili Prefecture	43.84 N, 81.15 E
July 2023	JMSD1	Dayou Town, Jimsar County, Changji Prefecture	43.50 N, 89.03 E
August 2023	JMSD2	Dayou Town, Jimsar County, Changji Prefecture	43.80 N, 89.04 E

**Table 2 insects-15-00559-t002:** Primers and their application in this study.

Gene	GenBank Accession	Primer Sequence (5′–3′)	Product Size (bp)	Application
*CYP9Z140*	XP_023030479.1	F: TAACGAGTTTAGCGTCAG	1881	Cloning
		R: CAATTGTTAATATGGAAGAC		
*CYP9AY1*	XP_023020330.1	F: TCGGTGGAATACCCATAT	1916	
		R: CAAACCAAATCCAAAACA		
*UGT321AP1*	XM_023173906.1	F: TCGAAACAGTGTTGGATATT	1663	
		R: AGTTTGACATGGCAACTTAG		
*CYP9Z140*		F: ACATGGCCCGAGGAATTGTA	157	qPCR
		R: TTTTCAACGGCAAGGACCAC		
*CYP9AY1*		F: CATTCGGCATTGGTCCAAGA	163	
		R: CCTTCTGGGCGCATATTGAA		
*UGT321AP1*		F: CATCAGGAAATGGCTACCGC R: AGACCCACAGCTATGCCTTT	189	
*RPL4*	EB761170	F: AAAGAAACGAGCATTGCCCTTCC	119	
		R: TTGTCGCTGACACTGTAGGGTTGA		
*EF1α*	EB754313	F: AAGGTTCCTTCAAGTATGCGTGGG	184	
		R: GCACAATCAGCTTGCGATGTACCA		
*CYP9Z140*		F: AGATCAGCAAACAGCCAGTAGTCAC	394	RNAi
		R: TATTAGCCCACAATGGATCAACATC		
*CYP9AY1*		F: TCGCAAATGATGTTATAGCTTCTTG	231	
		R: ATGGTACTATGGATGAGGTCGTGAA		
*UGT321AP1*		F: CGTCGCTGGTTAATCTCA R: GGGTGCGTAGGGTTGC	337	

**Table 3 insects-15-00559-t003:** Susceptibility to thiamethoxam of different populations of *L. decemlineata* adults in Xinjiang (2021–2023).

Year	Population	Slope ± SE	LD_50_ (µg/Beetle)/(95% FL)	Resistance Ratio
2021	URMQY	2.1167 ± 0.0528	0.0311 (0.0238–0.0408)	1.00
	QPQLZ	1.7757 ± 0.1442	0.2592 (0.1797–0.3741)	8.33
	JMSL	3.0319 ± 0.2467	0.2234 (0.1870–0.2669)	7.18
	URMQA	2.0484 ± 0.0546	0.0944 (0.0717–0.1243)	3.04
	ML	2.5969 ± 0.1200	0.0679 (0.0549–0.0842)	2.18
2022	URMQA	1.5184 ± 0.0336	0.2963 (0.1505–0.5834)	9.52
	JMST	2.6984 ± 0.1913	0.1006 (0.0823–0.1228)	3.23
2023	ML	1.0797 ± 0.0213	0.2309 (0.1057–0.5044)	7.42
	JMSD1	1.9694 ± 0.1116	0.2072 (0.1445–0.2970)	6.66
	URMQA	1.5521 ± 0.0887	0.1440 (0.0838–0.2083)	4.63
	JMSQ	3.5502 ± 0.1599	0.1310 (0.1068–0.1485)	4.21
	JMSD2	1.9202 ± 0.0577	0.0946 (0.0630–0.1422)	3.04
	QPQLB	1.3473 ± 0.0956	0.0690 (0.0355–0.1319)	2.22

**Table 4 insects-15-00559-t004:** Sequencing results of different populations of *L*. *decemlineata* ^a^.

Samples	Clean Reads	Clean Bases	GC Content (%)	Q30 (%)
URMQY	26,114,887	7,748,786,787	41.23	94.17
JMSL	24,582,542	7,263,561,882	40.52	93.39
QPQLZ	21,000,522	6,225,521,885	40.94	94.16

^a^ Clean reads: total number of pair-end reads of clean data. Clean bases: total number of bases. GC content: percentage of G and C bases in the DNA or RNA molecule. Q30%: percentage of bases with a mass value ≥ 30.

**Table 5 insects-15-00559-t005:** P450 and UGT genes upregulated significantly in transcriptomes of different groups of *L. decemlineata*.

Gene Function	Gene ID	URMQY vs. JMSL		URMQY vs. QPQLZ	
		log_2_FC	FDR Value	log_2_FC	FDR Value
CYP4C1-like	111503441	1.47	0.0018		
CYP12a5	111514589	1.64	0.0167		
CYP9e2-like	111508872			1.30	0.0035
CYP9e2-like	111518298			2.28	1.44 × 10^−17^
CYP9e2-like	111508919			1.56	5.11 × 10^−7^
CYP4V2-like	111510743			1.46	0.0453
CYP6a13	111506689			1.33	0.0004
CYP4c3-like	111504218			1.47	0.0132
UGT2B4-like	111517685	2.43	0.004		
UGT2B7-like	111518183			2.50	0.008

## Data Availability

All data presented in this study are available in this article.

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
