# Peer review of "Expression and Functional Analysis of Two Cytochrome P450 Monooxygenase Genes and a UDP-Glycosyltransferase Gene Linked with Thiamethoxam Resistance in the Colorado Potato Beetle"

_insects, 2024, doi:10.3390/insects15080559_

Round 1

Reviewer 1 Report

Comments and Suggestions for Authors

Author Response

Authors have conducted a significant study on resistance status to Thiamethoxam and mechanism of resistance in different wild population of Colorado Potato Beetle (CPB). Using transcriptome analysis, they have identified three genes in CPB which may be involve in metabolic resistance to Thiamethoxam. Then they have validated their findings with RT-PCR. Furthermore, they have functionally confirmed the involvement of these genes in Thiamethoxam resistance using RNA interference (RNAi) assays. This is very important and challenging work. This study can provide important insights into resistance development in agricultural pests such as CPB. Their findings will help in the development of successful CPB resistance management strategies. However, the manuscript requires some minor revisions, which are detailed below:

  1. Line no. 17: Add ‘After’ after ‘enhanced’

Thank you for pointing this out. We have revised it on line 20. 

  1. Line no. 28: Add ‘of resistance’ after low level

We have revised it on line 31 according to you. 

  1. Lines 57-58: Provide a reference for the statement: “In addition, research has shown that resistance to neonicotinoids was commonly related to the enhanced activity of detoxification enzymes, particularly cytochrome P450 monooxygenases (P450s).”

We have added references on line 62 according to you.

  1. Table 1: Correct the sampling dates to eliminate topological errors and add the coordinates of the sampling sites.

We have revised according to you.

  1. The sample size during expression analysis in different developmental stages are different; authors should explain logic behind this.

We have revised according to you on line 165-167.

  1. Line no 176: It world be better if authors would mentioned the concentration of primer and cDNA template (Working) in RT-qPCR.

We have revised according to you on line 189-190.

  1. Line no 178: denaturation temperature for RT-qPCR needs correction

Thank you for pointing this out. We have revised it on line 191. 

  1. Table no 3: authors are advised to represent this data in bar graph; it would be beneficial for readers to compare RR50 of same population over the years.

Thank you for your suggestion, but only a same population has been continuously monitored for three years, so the LD50s and RRs data are presented in table form.

  1. In Figure 6: UGT321AP1 expression in URMQA population is highly expressed in 2021 and 2022 but not in 2023 but RR50 in 2022 is very higher than 2021 & 2023; so, authors should explain why expression in 2021 is very despite having RR50 similar to 2023. Thank you for pointing this out. UGT321AP1 expression in URMQA population is highly expressed in 2021 and 2022 but not in 2023, whereas RR in 2022 is very higher than that in 2021 & 2023 with similar levels. On the one hand, expression of UGT321AP1 in the same population during different years may change with the resistance levels to thiamethoxam. On the other hand, the expression may also be affected by the different resistance background of population caused by the migration of individuals from other areas. So its expression level may not show a very good positive correlation with the resistance level of thiamethoxam.
  2. Line no 375 -376: Correct the typographical errors in the spelling of “first” and add “instar” after “second.” Also, correct the spelling of "Pupa" or "Pupae."

Thank you for pointing these out. We have revised according to you on line 389-390. 

  1. Line no 463-464: Provide a reference for: “However, the highest stage-specific expression of CYP6FV12 was and observed in 463 fourth-instar nymphs of B. odoriphaga and CYP303A1 had the highest expression in the ring 464 gland of D. melanogaster.

We have revised according to you on line 487.

Reviewer 2 Report

Comments and Suggestions for Authors

This manuscript identifies three genes (two CYPs and a UGT) that contribute to thiamethoxam resistance in CPB populations in China. The genes were found by RNA-seq of resistant, in comparison with sensitive, CPB populations and then their expression characterized by qPCR of various tissues and life stages of CPB, as well as in insecticide exposed and unexposed groups. Finally, RNAi silencing of each gene was used to demonstrate a role in thiamethoxam resistance. Overall, the work done is suitable for the goals of the study and the interpretation of the results is generally appropriate. There are some specific points that I might question and there are some aspects of the presentation that could be improved and these are summarized below.

Some parts of the manuscript are less clearly written. For example, the Simple Summary requires improvement. Line 14: this should be “differentially”, not “differently”. Line 17: …was significantly enhanced “upon” exposure to thiamethoxam. Lines 19-20: the findings reveal that these genes have roles in thiamethoxam resistance of L. decemlineata, not what the roles are.

Line 98: please indicate the potato variety used in the experiments.

Line 99-100: The authors say that “adults with same size and good growth were selected for subsequent experiments”. Since adult CPB are sexually dimorphic for size (females typically larger than males), selecting insects based on size could bias cohorts based on sex. The authors may want to elaborate on this point.

Table 1: date formats are not consistent.

Line 112 and subsequently: since the “treatment” is delivered at the start of the bioassay, it might be better to say “72 hr after treatment”.

Line 114: please clarify the meaning of “three as a repeat”.

Line 116: Since no comment is made, it seems that whole adults were used for RNA-seq? Some information regarding the RNA extraction methods used on the whole adults by Biomarker Technologies would be helpful.

Line 128: Blast2GO

Line 148: here is one instance where UGT is incorrectly written as UTG. This needs to be checked throughout the manuscript.

Line 159: “survivors” not “survival”.

Line 178: The denaturation step of the cycles is presumably 90 degrees.

Line 181: this is EF1”alpha” not EF1a.

Line 203: “formula” not “formular”.

Lines 207-209: This sentence needs to be rewritten.

Tables 3 and 4: keep sample naming consistent between tables.

Lines 261-262: “transcriptomes of different groups of L. decemlineata” not “different groups of Leptinotarsa decemlineata transcriptome”.

Lines 264-267 and Fig 3: it would be helpful to know here which gene ID's in Table 5 correlate to CYP9Z140, CYP9AY1, UGT324BR1 and UGT321AP1.

Lines 266-267: Given the increase in UGT gene expression measured by qPCR was greater than 10x higher than when measured from RNA-sequencing data, this is not really "consistent"?

Fig 4 legend: Please provide a better description of panel A. There are no red arrows that I can see and the characters present in the consensus line here and in Fig 5A need some explanation.

Line 350: “exposed” not “exposure”.

Fig 8: This seems contradictory between panels B and C. Panel B suggests UGT321AP1 is mainly expressed in MT and FB (which are in the abdomen), yet panel C results suggest it is mainly expressed in head and thorax. The authors might want to comment on this.

Line 376: second-instar larva.

Line 386: based on the Materials and Methods, this ratio should be “1:1:1”.

Line 391: should be CYP9Z140.

Line 401: This is not clear what is meant here. Does it mean 0.0196 ug/adult was the LD50 for thiamethoxam in QPQLZ individuals in 2010?

Line 403: It is also not made clear here that the 4.3 fold RR reported by Shi et al. was obtained using insects from 2018/19 seasons.

Lines 409-410: This sentence seems reversed. Long-term application of thiamethoxam in JMS populations would explain INCREASED levels of tolerance, not decreased levels.

Line 484: were found “to be” upregulated “in a” spinosad resistant population of CPB…

Comments on the Quality of English Language

Content related typographical errors have been noted above. Minor problems with grammar can be edited by the publisher.

Reviewer 3 Report

Comments and Suggestions for Authors

This study first collected populations from Xinjiang province and monitoring the resistance to thiamethoxam, and then using transcriptome sequencing to identified the differentially expressed genes, RT-qPCR and RNAi were used to found and confirmed the CYP9Z140, CYP9AY1, and UGT321AP1 were related to thiamethoxam resistance. These results were benefit for the development of CPB management strategies. Basically, this manuscript was well organized and written. Some comments or suggestions as follow.

1. Please use the full name of genus name when its first appearance, followed by an abbreviation, for example, Leptinotarsa decemlineata should be abbreviation as “L. decemlineata” in tables and figures legends, sub-titles. Also, some place Tribolium should be abbreviate, please check through the manuscript.

2. Page 3, line 97, “in Xinjiang from June to July” should be “in Xinjiang from June to August”, because sampling date in Table 1 include August.

3. Page 3, 2.2. Bioassay, the adults for bioassay were catched from field or catched the insect from field and rearing for one generation in the lab?

4. Page 4, line 129, delete “data” in “data database”

5. Page 4, line 139, what’s mean of “first-fourth-instar larvae”? does it means first to fourth instar larvae?

6. Page 4, lines 154-156, “In addition, we collected 30 eggs (E), 30 1st-instar larvae (L1), 20 2nd instar larvae (L2), ten 3rd-instar larvae (L3), as well as three 4th-instar larvae (L4), pupae (P) and adults (A), respectively,”. “1st, 2nd, 3rd, 4th” should be “1st, 2nd, 3rd, 4th”, the number of eggs, 1st instar larvae, 2nd instar larvae use Arabic numerals, why 3rd instar larvae use English number (ten)? Please clear the number of pupae and adults used.

7. Page 10, line 259, gene ID of 111508919 in table 5, the fold change is 1.56, not as you said >2.

8. Page 10, line 264, please mention the CYP9Z140 and CYP9AY1 corresponding to which gene ID in table 5, CYP9Z1401 is 11518298 or 111508919?

9. Page 11, result 3.3, the gene names used in this paragraph seems use different font size. Also, please check lines 315, 321, 324, 341, 342, 383, 385, 387, 392.

10. Page 11, lines 279-280, both two P450 genes contained a 1,578 bp ORF and 525 aa?

11. Figure 6 and 8, three bars are not easy distinguish from each other, especially for CYP9AY1 and UGT321AP1, suggest change the bar of UGT321AP1 to white, then three bars are black, gray, and white.

12. Page 16, lines 384 and 386, added a comma before “and UGT321AP”

13. Page 16, lines 385-386, “mixture targeting three genes at a ratio of 1:1:11”? but in page 6, line 194, the ratio is 1:1:1. In fact, only report the results in the results section, do not repeat the methods again.

14. Page 16, line 387, “F=143.4401”, please unified the significant digits in whole manuscript.

Comments on the Quality of English Language

It's well written and only some place need refine.

Reviewer 4 Report

Comments and Suggestions for Authors

This study investigated whether the differentially expressed P450 genes CYP9Z140 and CYP9AY1 and UGT gene UGT321AP1, found from our transcriptome results conferred re-sistance to thiamethoxam in L. decemlineata. Resistance monitoring showed that sampled ffeld populations of L. decemlineata adults collected from Urumqi City, Qapqal, Jimsar and Mulei County of Xinjiang in 2021-2023 developed low levels to thiamethoxam with resistance ratios ranging from 6.66- to 9.52-fold. Expression analyses indicated that CYP9Z140, CYP9AY1, and UGT321AP1 were  signiffcantly upregulated in thiamethoxam-resistant populations compared with susceptible population.    Overall, this work looks very interests and can be accept for publication before revision.

(1)  In the introduction section, I suggest author need tell us what's your purpose and why you did this work? and also, biopesticides based on RNAi took a major step forward with the first registration of a sprayable RNAi product, Ledprona, which was approved by the United States Environmental Protection Agency (EPA) on December 22, 2023, such big issue need included into introduction section. see the recently publication (Yan JJ, Nauen R, Reitz S, Alyokhin A, Zhang J, MotaSanchez D, Kim YY, Palli SR, Rondon SI, Nault BA, JuratFuentes JL, Crossley MS, Snyder WE, Gatehouse AMR, Zalucki MP, Tabashnik BE & Gao YL. The new kid on the block in insect pest management: sprayable RNAi goes commercial. Science China Life Science. 2024. https://doi.org/10.1007/s11427-024-2612-1).

(2)  In the discussion section, author need tell us why the RR still very low in China?

(3) Unfortunately, the English in the text is poorly written. The language and sentence structures of this manuscript are at many times incomprehensible. The paper needs rewriting and thorough language editing.

Comments on the Quality of English Language

 Unfortunately, the English in the text is poorly written. The language and sentence structures of this manuscript are at many times incomprehensible. The paper needs rewriting and thorough language editing.

Round 2

Reviewer 4 Report

Comments and Suggestions for Authors

looks much better and this version can be accepted for publication